# ProxSparse: Regularized Learning of Semi-Structured Sparsity Masks for Pretrained LLMs

Hongyi Liu [1] [*]   Rajarshi Saha [2]   Zhen Jia [2]   Youngsuk Park [2]   Jiaji Huang [2]   Shoham Sabach [2] [3]
Yu-Xiang Wang [2] [4]   George Karypis [2]

## Abstract

Large Language Models (LLMs) have demonstrated exceptional performance in natural language processing tasks, yet their massive size makes serving them inefficient and costly. Semi-structured pruning has emerged as an effective method for model acceleration, but existing approaches are suboptimal because they focus on local, layer-wise optimizations using heuristic rules, failing to leverage global feedback. We present **ProxSparse**, a learning-based framework for mask selection enabled by regularized optimization. ProxSparse transforms the rigid, non-differentiable mask selection process into a smoother optimization procedure, allowing gradual mask exploration with flexibility. ProxSparse does not involve additional weight updates once the mask is determined. Our extensive evaluations on 7 widely used models show that ProxSparse consistently outperforms previously proposed semi-structured mask selection methods with significant improvement, demonstrating the effectiveness of our learned approach towards semi-structured pruning. Code available here.

## 1. Introduction

Large Language Models (LLMs) have demonstrated strong performance across a wide range of natural language processing (NLP) tasks (Achiam et al., 2023; Wei et al., 2022). However, deploying and serving LLMs is not cost-efficient due to their massive size with billions of parameters (Frantar & Alistarh, 2023; Sui et al., 2025). To address the high computational demands and improve accessibility, various techniques have been proposed to make LLMs more efficient, such as model compression (Han et al., 2015; Frantar et al., 2022). By reducing memory footprint and accelerating computation, model compression significantly improves the feasibility and cost-effectiveness of deploying LLMs at scale (Yuan et al., 2024; Lin et al., 2024; Tseng et al., 2025; Ozkara et al., 2025; Wei et al., 2025).

Network pruning is commonly used to reduce model size and lower computation cost by removing unimportant parameters (Bai et al., 2024). Among various pruning patterns, semi-structured pruning (Mishra et al., 2021), or block-wise N:M sparsification, has emerged as a practical and effective approach for LLM compression (Sun et al., 2023; Fang et al., 2024). In this approach, only N non-zero elements are retained out of M consecutive elements within each parameter block. This semi-structured sparsity strikes a balance between model accuracy and hardware efficiency, and is well-supported by many hardware accelerators (Mishra et al., 2021), enabling efficient LLM serving.

Despite its advantages, finding an effective semi-structured mask for LLMs remains challenging. Pruning must follow per-block structural restriction, making efforts on other patterns hard to adopt. Additionally, extensive retraining after pruning is impractical due to LLMs' massive size (Ma et al., 2023; Chuang et al., 2024). Recent advances like Wanda (Sun et al., 2023) and SparseGPT (Frantar & Alistarh, 2023) improved semi-structured pruning using minimal resources with only hundreds of calibration samples, but still struggle to maintain optimal performance after pruning. We identify two main challenges in finding effective semi-structured masks: **1.** The heuristic rules used for mask selection cannot fully take advantage of the calibration dataset during pruning. Methods like SparseGPT and Wanda rely on the Hessian matrix and importance scores to select elements to prune, but these lightweight criteria fail to effectively leverage or learn from the calibration data. **2.** Both methods focus on solving a "local" optimization problem associated with individual layer, without considering the broader, end-to-end optimization across the entire model. In those methods, pruning is based on localized information within each layer, without considering the connections across layers. Thus they cannot benefit from the global feedback, limiting the effectiveness of the pruning method.

---

[*]Work done during internship at Amazon Web Service. [1]Rice University [2]Amazon Web Service [3]Technion [4]UCSD. Correspondence to: Hongyi L. <hongyi.liu@rice.edu>, Rajarshi S. <sahrajar@amazon.com>, Yu-Xiang W. <yuxiangw@ucsd.edu>.

*Proceedings of the $42^{nd}$ International Conference on Machine Learning*, Vancouver, Canada. PMLR 267, 2025. Copyright 2025 by the author(s).

We advocate a learning based solution for semi-structured mask selection that incorporates global feedback. We propose **ProxSparse**, which learns to discover semi-structured masks through an end-to-end optimization process, rather than solely relying on local, heuristic-based decisions. ProxSparse enables a finetuning-like procedure that learns the mask through only hundreds of calibration datasets with low resource utilization. The core of ProxSparse is the mask selection regularizer applied during learning, which transforms the rigid, non-differentiable mask-selection problem into a gradual search process. ProxSparse progressively enforces semi-structured sparsity and frozen weight constraints during training, and gradually shrinks unimportant weights to be pruned. ProxSparse does not involve additional weight updates after determining the mask. One challenge in regularized learning is the efficiency of the solver, as a slow solver makes end-to-end learning on LLMs impractical. To address this, we developed a fast solver using iterative soft-thresholding, enabling efficient end-to-end learning at LLM scale.

To comprehensively evaluate our method, we conducted extensive experiments on 7 widely used high-performance open-source models from four model families including Mistral (Jiang et al., 2023), Qwen (Yang et al., 2024), OpenLlama (Geng & Liu, 2023) and Llama (Touvron et al., 2023) family. The benchmarks cover language modeling and seven widely used natural language reasoning tasks. The results show that our regularized learning significantly outperforms baselines consistently accross all evaluated models, producing more effective pruning masks. Our contributions are summarized as follows:

- We propose to apply mask selection regularizer for end-to-end learning of semi-structured masks in LLMs. It allows gradual mask discovery with gradient feedback, enabling global optimization with flexibility, which leads to substantial improvements.
- We developed an efficient proximal gradient descent solver for the semi-structured sparsity regularizer. This method is 10x faster than gradient descent-based solvers and 100x faster than Interior Point Method (IPM) solvers, enabling end-to-end regularized learning at LLMs scale efficiently.
- Across all tested models, ProxSparse consistently improved perplexity (PPL) and accuracy on 7 commonsense reasoning tasks. It outperforms the previous SOTA pruning baselines at the same scale by up to 35% in PPL and 20% in zero-shot tasks, highlighting its effectiveness.

## 2. Preliminaries and Problem Setup

### 2.1. Large Language Model pruning

The massive size of LLMs has drawn attention to model compression to reduce serving overhead. Network pruning effectively removes redundant parameters, improving efficiency. In LLMs, pruning has proven effective (Bai et al., 2024; Frantar & Alistarh, 2023; Huang et al., 2024), and can be categorized into three classes based on granularity.

Structured pruning (Ma et al., 2023; Xia et al., 2023) removes entire substructures like neurons or attention heads, reducing computation without extra overhead. However, its rigidity and lack of flexibility often lead to significant performance loss, requiring additional retraining to recover accuracy (Ma et al., 2023; Xia et al., 2023). Unstructured pruning (Frankle & Carbin, 2018) effectively preserves model accuracy by selectively removing unimportant weights in a fine-grained, non-uniform manner. However, its irregular pruning pattern is hardware-unfriendly, causing inefficient memory access. Semi-structured (block-wise N:M) sparsity (Mishra et al., 2021) balances accuracy and efficiency by retaining N non-zero elements per M-sized block. Such patterns can be effectively leveraged by commercial hardwares for real speedup (Fang et al., 2024; Sun et al., 2023; Mishra et al., 2021), while maintaining flexibility to remove unimportant weights. This work focuses semi-structured pruning for LLMs, introducing an end-to-end regularized learning framework towards optimal mask selection.

### 2.2. Semi-Structured masks Selection for LLMs

Previous research has explored various mask-finding techniques for LLMs, with many showing success in semi-structured pruning. Here, we review the most advanced methods for semi-structured mask selection.

Magnitude pruning (Han et al., 2015) is a standard technique that removes individual weights based on their magnitudes with certain thresholds. Wanda (Sun et al., 2023) also avoids retraining or updating weights and introduces activation-aware pruning. The importance of each weight is evaluated using a per-output pruning criterion, where the weight magnitude is multiplied by its corresponding input activation using calibration data. SparseGPT (Frantar & Alistarh, 2023) leverages the Hessian matrix to calculate the weight importance and reconstruction errors with the calibration data. These pruning methods typically solve a local optimization problems, providing efficient and low-resource compression techniques (Ma et al., 2023; Frantar & Alistarh, 2023; Frantar et al., 2022; Sun et al., 2023).

On the other hand, learning-based solutions for pruning have been explored in previous works, particularly in vision tasks. The main challenge is the non-differentiable nature of mask selection, and techniques like Straight-Through Estimators (STE) (Bengio et al., 2013) have been proposed to overcome this. However, these methods typically require large-scale retraining, which is difficult for LLMs due to their enormous size. In our work, we propose to use the mask selection regularizer and efficiently identify the optimal mask in a

learned manner with only hundreds of calibration samples without extensive retraining. A recently proposed learning-based method, MaskLLM (Fang et al., 2024), introduces a large-scale learning-based approach ($\sim$100,000 samples) to learn pruning masks using Gumbel Softmax sampling. Our approach employs a different design and operates with $\sim$1000x smaller sample size ($\sim$100 samples). We consider MaskLLM complementary to our approach, as it focuses on the regime that learns with large-scale data samples. We provide more comparison and discussion in Sec. 4.3.4.

### 2.3. Problem setup

Let $W_0 \in \mathbb{R}^d$ be the pre-trained weights of the model and $\mathcal{L}(W)$ be the (population) loss function for the model with weight $W$. We say a $W \in \mathbb{R}^d$ is $2:4$-sparse if for every block of 4 parameters in $W$ only 2 are non-zero.

Our goal is to solve the pruning problem by finding an appropriate semi-structured sparse masks while keeping the weights of the pretrained model frozen.

We may express our task using the following stochastic optimization problem:

$$\min_M \quad \mathcal{L}(W_0 \odot M),$$
$$\text{s.t.} \quad M \in \{0,1\}^d, M \text{ is 2:4 sparse,} \tag{1}$$

where $\mathcal{L}$ denotes the loss, mask $M \in \{0,1\}^d$ denotes a Boolean-valued with the same shape as the frozen model weights $W_0$, and $\odot$ denotes element-wise multiplication. The problem is hard to solve because $\mathcal{L}$ is non-convex and the constraints are combinatorial. Moreover, we do not have access to $\mathcal{L}$ directly (since it's the *expected* loss). Instead, we have a small calibration dataset that we can stream through that gives us *stochastic* first-order (gradient) access, if we assume they are new data points drawn from the test-data distribution.

Given these constraints, our goal is not to *solve* (1), but rather to find efficient heuristics that work in practice. In Section 3, we propose our approach and highlight the interesting aspect of it. In Section 4, we thoroughly evaluate our method in semi-structured sparse pruning in a number of open-source LLM models.

## 3. Methodology

We introduce ProxSparse, a learning-based pruning method guided by a mask selection regularizer that generates high-quality semi-structured masks for efficient LLM serving. ProxSparse enables mask exploration in a global perspective by leveraging the gradient-based method, taking into account cross-layer connections with end-to-end feedback, rather than relying on localized, heuristic-based approaches for abrupt pruning. In this work, we focus specifically on

2:4 sparsity, and we discuss the extension to other sparsity patterns in Appendix G.

To address the challenges posed by the non-convex and non-differentiable nature of (1), our strategy for solving (1) involves (a) designing a relaxation of the problem with hard constraints into a (Lagrange) regularized form (b) developing a principled optimization algorithm for solving the relaxed problem, thereby facilitating the learning process.

### 3.1. Relaxation and Structure-inducing regularization

We start by rewriting (1) into an equivalent form:

$$\min_W \quad \mathcal{L}(W),$$
$$\text{s.t.} \quad W \text{ is 2:4 sparse,} \tag{2a}$$
$$\text{Mask}_W \odot (W - W_0) = 0, \tag{2b}$$

where $\text{Mask}_W$ selects the non-zero elements of $W$, $W_0$ denotes the original pretrained parameter weights.

This seemingly trivial reformulation changes the variables to optimize from a Boolean mask to a continuous weight vector which makes it more amenable to continuous optimization.

Next, we propose a relaxation of the two constraints (2a) and (2b) into a regularized form that gradually induces these structures:

$$\min_W \quad \mathcal{L}(W) + \lambda_1 \text{Reg}_{2:4}(W) + \lambda_2 \text{Reg}_{W_0}(W), \tag{3}$$

where $\text{Reg}_{2:4}(W)$ promotes the structured sparsity constraints and $\text{Reg}_{W_0}(W)$ penalizes the deviation away from the initial pretrained weight $W_0$.

$\text{Reg}_{2:4}$ decomposes into every 4-parameter block, where we apply the following regularizer (Kübler et al., 2025) to enforce the sparse pattern.

$$\begin{aligned} \text{Reg}_{2:4,\, w \in \mathbb{R}^4}(w) = &|w_1||w_2||w_3| + |w_2||w_3||w_4| \\ &+ |w_3||w_4||w_1| + |w_4||w_1||w_2|. \end{aligned} \tag{4}$$

**Proposition 1.** *The following statements hold true.*

1. *$Reg_{2:4,\, w \in \mathbb{R}^4}(w) = 0$ if and only if $w$ is 2:4 sparse.*

2. *$Reg_{2:4,\, w \in \mathbb{R}^4}(w)$ is invariant to permutation of the coordinates.*

3. *$Reg_{2:4,\, w \in \mathbb{R}^4}(w)$ is differentiable when restricting to the "active set" $\{i \in [4] \mid |w_i| > 0\}$.*

Observe that by the first property, if $\lambda_1 \to \infty$ the solution is guaranteed to be (2a). The non-smoothness of (4) ensures that it enjoys a "shrinkage" property (analogous to $\ell_1$-regularization for sparsity) such that it induces *exact* 2:4-sparsity even if $\lambda$ is not tending to $\infty$.

To promote the locality constraint (2b), we design the second regularizer as follows.

$$\text{Reg}_{W_0}(W) = \left\| \frac{W}{W_0 + \epsilon\,\text{sign}(W_0)} \odot (W - W_0) \right\|_F^2,$$

where the division is coordinate-wise and $\text{sign}(\cdot)$ outputs 1 when $\cdot \geq 0$ and 0 otherwise.

This regularizer can be viewed as a special weight decay towards $W_0$, but it imposes a stronger penalty for coordinates of $W$ that are larger and nearly no penalty for those coordinates that are nearly 0. $\epsilon\,\text{sign}(W_0)$ is added to avoid the numerical instability associated with (near)-0 division.

**Proposition 2.**   *1. $\text{Reg}_{W_0}(W) = 0$ if and only if $[W]_i = [W_0]_i$ for all coordinates $i$ s.t. $W_i \neq 0$.*

*2. $\text{Reg}_{W_0}(W) = \text{Reg}_{W_0[W \neq 0]}(W[W \neq 0])$.*

*3. $\text{Reg}_{W_0}(W)$ is continuously differentiable.*

Together with Proposition 1, we observe that the nullspace of the two regularizers together is the feasible region of the original problem, which allows us to optimize towards a solution that satisfies the original problem's constraints.

**Corollary 3.** *$\text{Reg}_{W_0}(W) = 0$ and $\text{Reg}_{2:4}(W) = 0$ if and only if $W$ satisfies (2a) and (2b).*

To say it differently, if $\lambda_1, \lambda_2 \to \infty$, the relaxed problem (3) is identical to the original problem (1). We encode the rigid and non-differentiable mask selection constraints into the learning objectives, enabling a learnable optimization process. Another benefit of transitioning from hard constraints to soft regularization is that it introduces "wiggling room", enabling flexibility during exploration. This allows the learning to make smoother, more informed pruning decisions with a larger exploration space, rather than making abrupt changes during optimization, which could cause early commitment to suboptimal state as we will show in experimental Section 4.3.2 later. The main challenge now lies in an effective solving algorithm for the regularizer with efficiency, which is crucial to facilitate end-to-end mask learning for LLMs with scale.

### 3.2. Proximal Gradient Descent for 2:4 Sparsity

To optimize (3), we propose to use the proximal gradient descent (Nesterov, 2013) — a popular method for solving composite optimization problems of the form $\min_x f(x) + h(x)$ where $f$ is differentiable but $h$ is not.

Proximal gradient descent iteratively updates $x$ by alternating between a gradient descent step on $f$ and a proximal operator (a generalization of "projection") on $h$:

$$y = x_t - \eta \nabla f(x_t), \tag{5a}$$

$$x_{t+1} = \arg\min_x \frac{1}{2}\|x - y\|^2 + h(x). \tag{5b}$$

---

**Algorithm 1** ProxSparse: Proximal Gradient Descent for End-to-End 2:4-Sparsity Pruning

---

1: **Input:** Initial pretrained weights $w_0$. Learning rate schedule $\eta_0, \eta_1, \ldots$. Stochastic gradient oracle $\mathcal{G}$ that takes $w$ and outputs $g$ such that $\mathbb{E}[g] = \nabla\mathcal{L}(w)$.
2: **for** $k = 0, 1, 2, \ldots$ **do**
3:     $g_k \leftarrow \mathcal{G}(w_k)$          $\triangleright$ SGD (or Adam) update.
4:     $V \leftarrow W_k - \eta_k(g_k + \lambda_2 \nabla\text{Reg}_{W_0}(W_k))$
5:     $W_{k+1} \leftarrow \arg\min_W \frac{1}{2}\|W - V\|^2 + \lambda_1\text{Reg}_{2:4}(W)$.
6: **end for**
7: **Output:** $W_0 \odot \text{Mask}_{\text{Proj}_{2:4}(W_k)}$.

---

In our problem, $f := \mathcal{L} + \lambda_2\text{Reg}_{W_0}$ and $h := \lambda_1\text{Reg}_{2:4}$. Pseudocode of this algorithm is given in Algorithm 1.

The main benefit of the proximal gradient approach is that it does not prematurely commit to a particular sparsity mask, or fix the weights at the initialization. Instead, the regularizers are soft constraints, allowing ample wiggling room around the rigid constraint set for the gradient descent-based algorithm to potentially jump out of a suboptimal local region, and thereby converge to a better qualifying solution.

One issue of not imposing the constraint is that the last iterate might not be feasible after the specified number of iterations. For those cases, we simply project the solution $W_k$ to a 2:4-sparse solution basing on magnitude and snap the surviving weights to $W_0$. All our experimental results are based on solutions that are exactly 2:4 sparse with weights unchanged from initialization.

### 3.3. Efficient Proximal Operator

An efficient solver for the proximal operator is essential for enabling end-to-end learning at LLM scale. Since $\mathcal{L}$ and $\text{Reg}_{W_0}$ are both differentiable, the efficient implementation of ProxSparse boils down to solving the proximal operator associated with $\text{Reg}_{2:4}$.

$$w^* = \arg\min_{w \in \mathbb{R}^4} \frac{1}{2}\|w - y\|^2 + \lambda\text{Reg}_{2:4}(w) \tag{6}$$

This is a non-convex optimization problem. Kübler et al. (2025) showed that it can be solved with three convex subproblems.

**Theorem 4** ((Kübler et al., 2025)). *To solve (6) for any $y \in \mathbb{R}^4$, it suffices to solve:*

$$\min_{w \in \mathbb{R}_+^4} \frac{1}{2}\|w - z\|^2 + \lambda\text{Reg}_{2:4}(w) \tag{7}$$

*where $z = \text{sorted}(|y|)$ is non-negative and sorted in descending order, i.e., $z_1 \geq z_2 \geq z_3 \geq z_4 \geq 0$. Moreover, the optimal solution to (7) must be one of the following three candidates:*

*1. "2-sparse solution"* $[z_1, z_2, 0, 0]$;

*2. "3-sparse solution"*, $[\dot{w}_1, \dot{w}_2, \dot{w}_3, 0]$

*3. "dense solution"* $[\ddot{w}_1, \ddot{w}_2, \ddot{w}_3, \ddot{w}_4]$

*where* $\dot{w} = \arg\min_{w \in \mathbb{R}^3_+} \{g_3(w) \ s.t. \ \nabla^2 g_3(w) \succeq 0\}$ *with*

$$g_3(w) := \frac{1}{2}\|w - z_{1:3}\|^2 + \lambda(w_1 w_2 + w_2 w_3 + w_3 w_1),$$

*and* $\ddot{w} = \arg\min_{w \in \mathbb{R}^4_+} \{g_4(w) \ s.t. \ \nabla^2 g_4(w) \succeq 0\}$ *with* $g_4(w)$ *being the objective function of* (7). *Meanwhile,* $\{w | \nabla^2 g_3(w) \succeq 0\}$ *and* $\{w | \nabla^2 g_4(w) \succeq 0\}$ *are convex sets, making the corresponding optimization problems convex.*

This result suggests that we can simply *enumerate* the three candidate solutions and return the one with the smallest objective value. Kübler et al. (2025) thus proposed to solve for the "3-sparse" and "dense" solutions using interior point method (IPM) with a log-determinant barrier function, leading to the EnumIPM algorithm, which optimally solves (7). However, EnumIPM incurs high computational cost (Table 1). A faster heuristic, EnumPGD, was introduced to replace IPM with projected gradient descent without imposing semidefinite constraints. While EnumPGD improves efficiency, it sacrifices provably guarantees.

We propose a new method based on alternating minimization (ALM) with convergence guarantees. The resulting EnumALM is even more efficient than EnumPGD (see Table 1 for an numerical comparison). Moreover, in all 20,000 experiments in Table 1, EnumALM provides more optimal solutions than those of EnumIPM. This enables us to scale up the proximal gradient method for handling LLMs with billions of parameters in practice. An example regularization paths is illustrated in Figure 3 in Appendix C.

Pseudocode for ALM and EnumALM are given in Algorithm 2 and 3 respectively. Algorithmically, ALM works by iterating over the coordinates of $w$ and minimizing $g_3$ or $g_4$ over the current coordinate while keeping other coordinates fixed. The solution of this one dimensional problem is soft-thresholding:

**Fact 5.** *Assume* $z \geq 0$, *the optimal solution to* $\min_{w \in \mathbb{R}_+} \frac{1}{2}(w - z)^2 + \alpha w$ *is* $w = \max\{z - \alpha, 0\}$.

Observe that soft-thresholding is commonly used in L1-regularized optimization for inducing (unstructured) sparsity. Our algorithm can thus be viewed as iterative soft-thresholding with adaptive chosen threshold that induces 2:4 structured sparsity rather than standard sparsity.

### 3.4. Convergence guarantees

Next, we study the convergence theory of ProxSparse. We first prove that the inner-loop Algorithm 2 always converges

---

**Algorithm 2** ALM: Alternating Minimization

1: **Input:** $z \in \mathbb{R}^4$ (sorted, nonnegative), parameter $\lambda$, tolerance $\epsilon$, desired sparsity-level $S = 3$ or 4.
2: Initialize $w' = 0, w = 0$,
3: $w_{1:S} \leftarrow z_{1:S}$
4: **while** $\|w' - w\| > \epsilon$ **do**
5:     **for** $i \in \{1, ..., S\}$ **do**
6:         $w_i \leftarrow \max\left\{z_i - \lambda \sum_{\substack{j,k \in [4]\backslash\{i\} \\ j \neq k}} w_j w_k, 0\right\}$
7:                   ▷ This is soft-thresholding operator
8:     **end for**
9:     $w' \leftarrow w$
10: **end while**
11: **Output:** $w$

---

**Algorithm 3** EnumALM for solving (6)

1: **Input:** $y \in \mathbb{R}^4$, parameter $\lambda$, tolerance $\epsilon$
2: $s \leftarrow \text{sign}(y)$             ▷ elementwise
3: $z, \text{idx} \leftarrow \text{sort}(|y|, \text{'descending'})$ ▷ idx is reverse index.
4: $\tilde{w} \leftarrow [z_1, z_2, 0, 0]$        ▷ 2-sparse solution.
5: $\dot{w} = \text{ALM}(z, \lambda, \epsilon, S = 3)$    ▷ 3-sparse solution.
6: $\ddot{w} = \text{ALM}(z, \lambda, \epsilon, S = 4)$      ▷ dense solution.
7: $w \leftarrow \arg\min_{w \in \{\tilde{w}, \dot{w}, \ddot{w}\}} \frac{1}{2}\|w - z\|^2 + \lambda \text{Reg}_{2:4}(w)$
8: **Output:** $s \odot w[\text{idx}]$    ▷ $\odot$ is elementwise product

---

| | EnumIPM | EnumPGD | EnumALM (ours) |
|---|---|---|---|
| Total runtime (sec) | 561.70 | 43.31 | 8.52 |
| Max suboptimality | $10^{-13}$ | $10^{-6}$ | $< 10^{-13}$ |

*Table 1.* Comparison of the runtime and accuracy of solvers of (6) for solving 100 randomly generated problem instances, each with 200 different choices of $\lambda$. The second row shows the worst-case suboptimality. IPM is guaranteed to give the optimal solution up-to a tolerance parameter of $10^{-13}$. ALM achieves better objective value in all experiments than IPM, while GD occasionally gives solutions with slightly suboptimal objective values.

to a critical point. Then we will argue that if Algorithm 3 returns the correct solution (they do in all our experiments!), then under mild assumptions on training loss $\mathcal{L}$ and boundedness of the parameters $W_k$, the outer-loop Algorithm 1 also converges to a stationary point. The proofs of both propositions below are deferred to Appendix A.

**Proposition 6** (Convergence of ALM). *When* $\epsilon > 0$, *Algorithm 2 halts with no more than* $3\lambda\|z\|^3/\epsilon^2$ *iterations. Also, at the limit* $\epsilon \to 0$, *the output of Algorithm 2 converges to a critical point of* $g_3$ *when* $S = 3$ *(or of* $g_4$ *when* $S = 4$*).*

**Proposition 7** (Convergence of ProxSparse). *Assume* $\mathcal{L}$ *is continuously differentiable, and that there exists* $B > 0$ *such that* $\|W_t\| \leq B$ *for all* $t = 1, 2, 3, ....$ *Then Algorithm 1 converges to a critical point in the sense of the limiting subdifferential of the regularized objective function of* (3) *(see (Rockafellar & Wets, 1998)).*

# 4. Empirical evaluation

In this section, we provide comprehensive evaluations of ProxSparse by addressing the following research questions: **1. End-to-end performance:** how does ProxSparse compare to other state-of-the-art pruning methods? **2. The in-depth analysis of mask selection regularizer:** how does the regularizer contribute to finding the effective mask? **3. Efficiency benefit:** does sparsified models produced by ProxSparse improve efficiency?

## 4.1. Models, tasks and baselines

We evaluated ProxSparse on four most advanced and widely used open-source LLM families: Mistral (Jiang et al., 2023), Qwen (Yang et al., 2024), OpenLlama (Geng & Liu, 2023) and an Llama (Touvron et al., 2023) family. The specific models used in our experiments include Mistral-v0.1-7b, Mistral-v0.3-7b, Qwen2.5-14b, OpenLlama-7b-v2, Llama-2-7b, Llama-2-13b and Llama-3.1-8b.

We assess the performance of pruned models from different pruning mechanisms on both zero-shot tasks and language modeling. For calibration, we followed Wanda (Sun et al., 2023) and SparseGPT (Frantar & Alistarh, 2023) to utilize the C4 (Raffel et al., 2020) dataset for calibration. Zero-shot performance was evaluated with the EleutherAI LM-Eval-Harness (Gao et al., 2024) on seven widely used tasks (Liu et al., 2024), while Wikitext (Merity et al., 2016) perplexity (PPL) was used as the language modeling metric, consistent with previous evaluation protocol (Sun et al., 2023; Frantar & Alistarh, 2023). The experiments use 400 data samples for calibration unless specified, with consistent counts across baselines for fair comparison. We discuss MaskLLM and present ablation studies on mask effectiveness with regards to calibration sample size in Section 4.3.4. Comparisons on additional pruning mechanisms (ADMMPrune (Boža, 2024), OWL (Yin et al., 2023) and AlphaPrune (Lu et al., 2024)) are further detailed in Appendix E. For hyperparameters and configurations, we detail them in Appendix B. Our experiments were done on Nvidia A100 GPUs.

## 4.2. End to end performance evaluation

We first present end-to-end performance comparison against other baselines that enforce 2:4 sparsity: magnitude pruning (Han et al., 2015), SparseGPT (Frantar & Alistarh, 2023), and Wanda (Sun et al., 2023). Table 2 presents Wikitext PPL and performance on seven widely used zero-shot reasoning tasks. Overall, ProxSparse consistently outperforms all baselines across tested models.

**Language modeling** We first evaluate language modeling. ProxSparse surpasses magnitude pruning and outperforms Wanda, the SOTA mechanism without weight updates at the same scale. More specifically, ProxSparse achieves a PPL of 9.91 vs. Wanda's 13.81 on OpenLlama-7b-v2 with 28% improvement. Similarly, ProxSparse achieves a PPL of 8.51 on Llama-2-7b, compared to Wanda's 11.42, reflecting a 35% improvement. In the Llama-3.1-8b experiments shown in Table 9, ProxSparse reduces PPL from Wanda's 20.91 to 13.63. Compare to the Llama-2-7b model, Llama-3.1-8b have more information encoded in the model weights as much larger training corpus was used during pretraining. This significant performance improvement highlights the potential of ProxSparse's effectiveness in handling dense model pruning mask selection. Even when compared to SparseGPT, which updates the weights to minimize error, ProxSparse still outperforms it by up to an 18% margin, as demonstrated in the Llama-2-7b experiments. In summary, across different models, ProxSparse consistently achieves better PPL with a significant gap compared to other baselines.

**Zero-shot Task Performance** We present the performance analysis on seven widely used zero-shot natural language reasoning tasks. Consistent with the language modeling results, ProxSparse significantly outperforms both magnitude pruning and Wanda. In the Mistral-v0.1-7B experiments, ProxSparse achieved an average accuracy of 52.7%, compared to Wanda's 44.1%, marking a 20% improvement in performance. Even with weight updates in SparseGPT, ProxSparse consistently achieves higher accuracy. Similar trends hold for Qwen2.5-14b and other models as well. This highlights ProxSparse's effectiveness in finding an optimal semi-structured mask to maintain superior performance, even compared to pruning methods with weight reconstruction for error reduction.

**Analysis of Better Performance** ProxSparse consistently outperforms all baselines across evaluated models. Its advantage stems from the global feedback in mask exploration, which enables ProxSparse to overcome localized constraints. By optimizing in an end-to-end manner, ProxSparse achieves superior performance gains.

## 4.3. Deep dive into the regularizing mechanism

This section explores the core properties of the mask selection regularizer. The regularizer relaxes rigid mask selection constraints into differentiable optimization for end-to-end learning. In the meantime, its added flexibility with "wiggling room" enhances exploration for better convergence. We ask the question: how does this flexibility aid in exploring the optimal mask during optimization?

### 4.3.1. HARD CONSTRAINT V.S. SOFT REGULARIZATION

To showcase the effectiveness of soft regularization with flexibility, we compare it with strict constraints. Unlike

*Table 2.* Experimental results on Wikitext perplexity (PPL) and 7 commonly used zero-shot natural language reasoning tasks comparing **ProxSparse** to 3 other baselines on 7 widely used LLMs (Llama-3.1-8b results are deferred to Table 9). **Bold** indicates the best pruning performance, while *italic* represents the original unpruned performance. SparseGPT updates weights to minimize reconstruction error, while the other methods keep retained weights frozen. ProxSparse consistently yields better results compared to all other baselines.

| | Weight Update | Wikitext PPL | ARC-C | ARC-E | SIQA | HellaSwag | OBQA | PIQA | TruthfulQA | AVG |
|---|---|---|---|---|---|---|---|---|---|---|
| Mistral-v0.1-7b | - | *4.91* | *0.503* | *0.809* | *0.467* | *0.612* | *0.324* | *0.806* | *0.354* | *0.554* |
| magnitude | ✗ | 14.18 | 0.310 | 0.666 | 0.417 | 0.488 | 0.204 | 0.732 | 0.314 | 0.447 |
| SparseGPT | ✓ | 9.43 | 0.345 | 0.684 | 0.418 | 0.469 | **0.240** | 0.730 | 0.316 | 0.501 |
| Wanda | ✗ | 11.49 | 0.336 | 0.665 | 0.408 | 0.444 | 0.214 | 0.716 | 0.307 | 0.441 |
| ProxSparse | ✗ | **8.92** | **0.362** | **0.698** | **0.428** | **0.525** | 0.232 | **0.756** | **0.350** | **0.527** |
| Mistral-v0.3-7b | - | *4.95* | *0.490* | *0.797* | *0.458* | *0.609* | *0.336* | *0.803* | *0.353* | *0.549* |
| magnitude | ✗ | 13.52 | 0.332 | 0.665 | 0.413 | 0.488 | 0.226 | 0.738 | 0.309 | 0.453 |
| SparseGPT | ✓ | 9.23 | 0.353 | 0.687 | 0.421 | 0.470 | **0.248** | 0.733 | 0.308 | 0.458 |
| Wanda | ✗ | 10.97 | 0.311 | 0.648 | 0.408 | 0.442 | 0.206 | 0.716 | 0.300 | 0.433 |
| ProxSparse | ✗ | **8.68** | **0.362** | **0.697** | **0.429** | **0.525** | 0.242 | **0.751** | **0.321** | **0.475** |
| Qwen2.5-14B | - | *4.93* | *0.56* | *0.822* | *0.554* | *0.634* | *0.342* | *0.814* | *0.493* | *0.602* |
| magnitude | ✗ | 48.87 | 0.359 | 0.638 | 0.405 | 0.418 | 0.256 | 0.680 | 0.356 | 0.444 |
| SparseGPT | ✓ | **9.19** | 0.405 | 0.750 | **0.476** | 0.512 | **0.296** | 0.753 | 0.367 | 0.507 |
| Wanda | ✗ | 11.69 | 0.389 | 0.729 | 0.440 | 0.491 | 0.286 | 0.740 | 0.331 | 0.485 |
| ProxSparse | ✗ | 9.28 | **0.456** | **0.772** | 0.456 | **0.535** | 0.290 | **0.756** | **0.406** | **0.525** |
| OpenLlama-7b-v2 | - | *6.48* | *0.387* | *0.725* | *0.441* | *0.557* | *0.296* | *0.789* | *0.336* | *0.504* |
| magnitude | ✗ | 36.15 | 0.230 | 0.498 | 0.380 | 0.360 | 0.162 | 0.683 | 0.306 | 0.374 |
| SparseGPT | ✓ | 11.35 | 0.278 | 0.602 | 0.412 | 0.428 | 0.214 | 0.713 | 0.301 | 0.420 |
| Wanda | ✗ | 13.81 | 0.261 | 0.575 | 0.409 | 0.409 | 0.196 | 0.703 | **0.310** | 0.409 |
| ProxSparse | ✗ | **9.91** | **0.281** | **0.616** | **0.415** | **0.472** | **0.236** | **0.720** | 0.299 | **0.434** |
| Llama-2-7b | - | *5.12* | *0.433* | *0.763* | *0.461* | *0.571* | *0.314* | *0.781* | *0.321* | *0.521* |
| magnitude | ✗ | 54.74 | 0.301 | 0.618 | 0.411 | 0.454 | 0.216 | 0.701 | 0.322 | 0.432 |
| SparseGPT | ✓ | 10.30 | 0.326 | 0.655 | **0.412** | 0.435 | 0.246 | 0.713 | 0.304 | 0.441 |
| Wanda | ✗ | 11.42 | 0.311 | 0.623 | 0.403 | 0.413 | **0.248** | 0.706 | 0.305 | 0.430 |
| ProxSparse | ✗ | **8.51** | **0.331** | **0.656** | 0.407 | **0.478** | 0.242 | **0.716** | **0.328** | **0.452** |
| Llama-2-13b | - | *4.57* | *0.485* | *0.794* | *0.473* | *0.601* | *0.352* | *0.791* | *0.314* | *0.544* |
| magnitude | ✗ | 8.32 | 0.319 | 0.623 | 0.408 | 0.501 | 0.232 | 0.717 | 0.309 | 0.444 |
| SparseGPT | ✓ | 8.14 | 0.378 | 0.714 | **0.437** | 0.478 | 0.282 | 0.735 | 0.296 | 0.473 |
| Wanda | ✗ | 8.35 | 0.340 | 0.683 | 0.424 | 0.464 | 0.246 | **0.739** | 0.292 | 0.455 |
| ProxSparse | ✗ | **6.61** | **0.383** | **0.720** | 0.427 | **0.532** | **0.288** | 0.723 | **0.319** | **0.486** |

the gradually sparse regularizer, projected gradient descent (PGD) imposes hard thresholding during optimization. We conducted four experiments to evaluate both regularizers for mask selection, testing each with both soft and hard constraints, as shown in Table 3. In proximal gradient descent, "hard sparsity constraints" in the table enforce zeroing two of every four weights after each update, ensuring rigid 2:4 structural sparsity. "Hard frozen weights" reset the two largest-magnitude weights to their original values, enforcing strict objectives for mask selection. With the relaxed regularizer, weights gradually shrink towards the 2:4 pattern (shown in Figure 4(a)), while the retained weights are encouraged to approximate their original values. This relax-

*Table 3.* Wikitext PPL under hard/soft constraints. Relaxing mask selection constraints improves performance over hard thresholding. Bold indicates the best result.

| | Both with relaxation | Fronzen weight relaxation | Sparsity constraints relaxation | Both with hard Constraints |
|---|---|---|---|---|
| Mistral-v0.3-7b | **8.68** | 13.23 | 11.24 | 13.6 |
| OpenLlama-7b-v2 | **9.91** | 34 | 33.07 | 35.28 |

ation meets both objectives under more flexible constraints. Table 3 indicates that hard constraints performs worst, while relaxed constraints enhance performance. Fully regularizing both semi-structured and frozen weight constraints maximizes flexibility, achieving the best results.

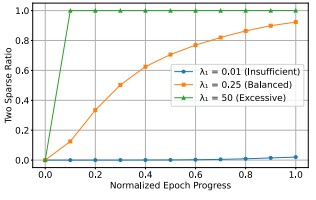
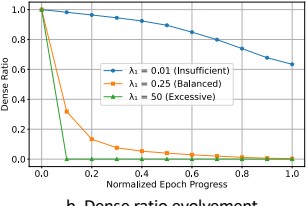
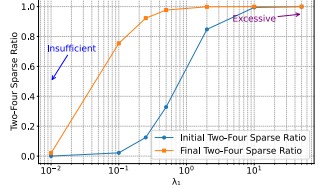
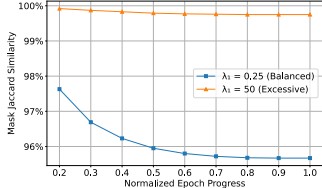

| a. Two-four sparse ratio evolvement | b. Dense ratio evolvement | c. Initial/final two-four sparse ratio vs. $\lambda_1$ | b. Mask similarity evolvement |

*Figure 1.* Evolution of sparsity ratio on Llama-2-7b based on the degree of regularization. (a) Evolution of the 2:4 sparsity ratio over learning progress, where an insufficient regularization degree leads to under-learning. (b) With a larger $\lambda_1$ parameters shrink more quickly towards 2:4 sparsity, resulting in early commitment to a suboptimal mask. (c) Comparison of the 2:4 sparse block ratios at early (0.1 epochs) and final stages of learning. (d) Mask similarity between the final mask and the early mask obtained after 10% epochs of learning. An excessively large $\lambda_1$ results in premature mask commitment, causing mask selection to stagnate and hindering optimal mask discovery.

*Table 4.* Wikitext PPL across $\lambda_1$. Optimal performance occurs at balanced regularization. Bold indicates the best performance.

| $\lambda_1$ | 0.001 | 0.01 | 0.1 | 0.25 | 0.5 | 2 | 10 | 50 | inf |
|---|---|---|---|---|---|---|---|---|---|
| Mistral-v0.3-7b | 14.19 | 9.66 | 9.04 | 8.69 | **8.68** | 8.82 | 9.32 | 10.94 | 11.33 |
| $\lambda_1$ | 0.001 | 0.1 | 0.25 | 1 | 2 | 5 | 10 | 50 | inf |
| OpenLlama-7b-v2 | 25.23 | 11.25 | 10.89 | **9.91** | 10.13 | 10.97 | 12.11 | 23.29 | 33.09 |

*Table 5.* Wikitext PPL across $\lambda_2$. A broad optimal plateau suggests that performance remains stable across a robust range of $\lambda_2$.

| $\lambda_2$ | 0 | 0.5 | 2 | 5 | 20 | 100 | 2500 | inf |
|---|---|---|---|---|---|---|---|---|
| Mistral-v0.3-7b | 8.68 | 8.88 | 8.9 | 8.82 | 8.99 | 8.85 | 9.11 | 13.23 |
| $\lambda_2$ | 0 | 0.5 | 2 | 20 | 100 | 500 | 2500 | inf |
| OpenLlama-7b-v2 | 9.91 | 9.92 | 9.96 | 10.12 | 10.41 | 10.9 | 12.38 | 34 |

### 4.3.2. THE SPARSITY PATTERN ENFORCER

In the following sections, we analyze the contribution of each regularizer individually, starting with the sparsity pattern regularizer, which encourages 2:4 sparsity. The regularizer coefficient, $\lambda_1$, controls the strength of regularization: higher values enforce more aggressive parameter shrinkage, approaching a harder projection with less flexibility. To isolate the effect of regularization, we only study the semi-structured regularizer in this analysis. We examine how varying its strength impacts mask learning. As shown in Table 4, optimal mask selection occurs at a balance between gradual and aggressive regularization—smaller values lead to conservative mask evolution, while larger values impose stricter constraints, both reducing performance.

To better understand this phenomenon, we analyze the regularizer's impact in detail. We show the evolution of 2:4 sparsity across different $\lambda_1$ values for Llama-2-7b (Figures 1) and OpenLlama-7b-v2 (Figure 4 in Appendix F) with consistent trend. We use Llama-2-7b as the example. if $\lambda_1$ is too low, the model remains largely dense, as shown in Figures 4(a), (b) and (c). This suggests under-learning, where unimportant weights are not fully recognized by the end of learning, resulting in incomplete mask selection. Conversely, a high $\lambda_1$ value leads to early commitment to a specific mask. In Figure 4(d), the yellow line shows similarity to the "early mask" obtained after just 10% of learning. The final mask retains ~99.5% similarity to the early one, indicating stalled optimization. In between, a balanced strength allows flexible mask exploration that avoids premature commitment, while also enabling more effective learning than excessively low values.

### 4.3.3. THE FROZEN WEIGHT RETENTION ENFORCER

We analyze the impact of the frozen weight regularizer, with its strength controlled by $\lambda_2$. Using the optimal $\lambda_1$ from previous analyses, we vary $\lambda_2$ to assess its effect. Interestingly, Table 5 shows a broad optimal performance plateau, suggesting that a robust range of $\lambda_2$ values can be applied without significantly affecting performance.

We further plot the evolution of the relative norm gap across $\lambda_2$ in Figure 2. This gap quantifies the difference in norm between the learned model and the original model, with the mask applied. It assesses how closely retained weights preserve their original values. We see that even without the regularizer, the relative norm gap stays ~20%. Adding the regularizer incurs small impact until the strength reaches a high extent. This may result from implicit frozen weight constraints imposed by the sparsity pattern regularizer, which we leave it to future investigation. As $\lambda_2$ increases toward infinity, strict projection enforcement degrades performance, aligns with previous finding and reinforces the need for gradual and flexible mask optimization.

### 4.3.4. PERFORMANCE EVOLUTION WITH VARYING NUMBERS OF CALIBRATION SAMPLES

Our method enables effective semi-structured mask selection with only hundreds of samples. Here we analyze performance based on the number of calibration samples with OpenLlama-7b-v2 and Llama-2-7b. We compare our results with MaskLLM, SparseGPT, and Wanda using 100, 200, and 400 samples. MaskLLM struggles with small sample sizes, making it a complementary method to ours in large-scale learning. We use the statistics reported in the

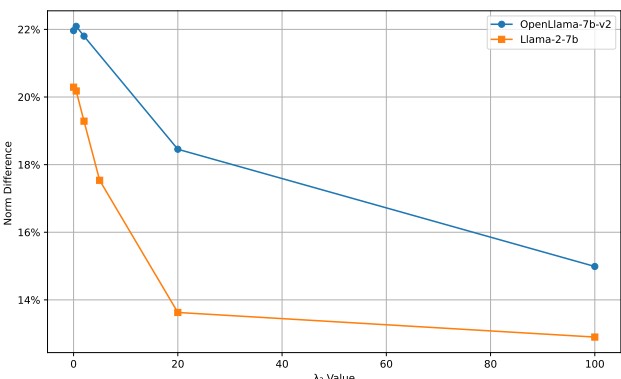

*Figure 2.* The relative norm difference over different $\lambda_2$. The relative norm gap measures how closely retained weights match their original values post-training, with the semi-structured mask applied. The relative norm remained low (∼20%) with minimal change until a high lambda value was applied.

*Table 6.* Wikitext PPL across calibration sample sizes. ProxSparse outperformed all methods, with performance slightly improved as sample size increased, confirming its effectiveness in optimal mask learning. Bold indicates the best performance.

| OpenLlama-7b-v2 | 100 | 200 | 400 | Llama-2-7b | 100 | 200 | 400 |
|---|---|---|---|---|---|---|---|
| MaskLLM | - | - | - | MaskLLM | > 13 | > 13 | > 11 |
| SparseGPT | 11.581 | 11.478 | 11.35 | SparseGPT | 10.36 | 10.32 | 10.298 |
| Wanda | 13.854 | 13.828 | 13.814 | Wanda | 11.46 | 11.45 | 11.42 |
| ProxSparse | **10.39** | **10.09** | **9.91** | ProxSparse | **9.24** | **8.99** | **8.51** |

MaskLLM paper (Fang et al., 2024). As shown in Table 6, MaskLLM performed worst on Llama-2-7b with low sample sizes, likely due to its reliance on extensive training for effective masks. SparseGPT and Wanda showed minimal improvement with increased calibration samples, consistent with previous observations (Fang et al., 2024; Sun et al., 2023). ProxSparse achieved the best across these sample sizes, with slight performance gains as samples increased within our target range. This confirms the effectiveness of our method in learning towards an optimal mask for semi-structured sparsity.

### 4.4. Improved efficiency during inference

Finally, we evaluate the efficiency metrics of the sparsified model produced by ProxSparse. We present wall-clock inference speedup and memory footprint improvements for the 2:4 semi-structured sparsified model induced by ProxSparse. Our experiments are conducted on Nvidia A100 GPUs. We utilize the Nvidia CUTLASS library as the underlying implementation for 2:4 semi-structured sparse operations.

#### 4.4.1. INFERENCE SPEEDUP

We follow the evaluation setup of previous work (Frantar & Alistarh, 2023; Sun et al., 2023) and measure the latency

*Table 7.* Speedup and memory utilization improvements achieved by ProxSparse induced 2:4 sparsity models(left: speedup, right: memory reduction). ProxSparse delivers a 1.3x–1.35x speedup for matrix multiplication and a 1.26x end-to-end inference speedup on the Mistral-v0.3-7b model. Additionally, ProxSparse reduces memory consumption by 29.5%–37.3% across different models, demonstrating its efficiency in both computation and memory utilization.

| Module name | Speedup ratio | Model family | Memory gain |
|---|---|---|---|
| self_attn q/k/v/o | 1.35x | Openllama_7b_v2 | 70.50% |
| mlp up/down/gate | 1.30x | Qwen2.5-14b | 67.50% |
| End-to-end inference | 1.26x | Mistral-v0.3-7b | 62.70% |

of matrix multiplication in linear layers. The results of Mistral-v0.3-7b (batch size of 1) are presented in Table 7. As shown in the table, 2:4 semi-structured sparsity induced by ProxSparse provides significant inference speedup for linear layers in LLMs, achieving an average speedup gains of 1.3 to 1.35. Additionally, we measured the end-to-end inference wall-clock speedup and observed a 1.26x speedup, consistent with other sparsification methods evaluated in our experiments. We emphasize that the inference speedup is not specific to our pruning method but rather a result of the inherent computational efficiency enabled by semi-structured sparsity.

#### 4.4.2. MEMORY FOOTPRINT IMPROVEMENTS

Next, we evaluate the memory footprint reductions achieved by ProxSparse-sparsified models. The results of peak memory utilization during inference time (batch size = 1) for different models are presented in Table 7. ProxSparse reduces peak memory usage by 29.5% to 37.3%, demonstrating significant memory savings with 2:4 sparsification. The exact reduction varies across different model architectures due to differences in model weight sizes, which influence activation sizes and ultimately affect peak memory consumption. Overall, ProxSparse effectively reduces memory footprint during LLM inference, highlighting the system benefits of the 2:4 sparse operation.

## 5. Conclusion

LLMs excel in natural language processing tasks and downstreaming tasks. However, they suffer from high computational costs due to the enormous parameter sizes. Semi-structured sparsity can improve inference efficiency, though it remains challenging due to the structural constraints during pruning. We propose a learning-based method with regularized optimization, progressively explores optimal mask through end-to-end gradient feedback. Extensive evaluation shows that ProxSparse significantly outperforms previous methods, enabling better accuracy for LLM pruning, making model deployment more cost-effective.

## Impact statement

This paper presents work with the goal is to advance the field of machine learning. There are many potential societal consequences of our work, none which we feel must be specifically highlighted here.

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

# A. Proofs of Technical Results

*Proof of Proposition 1.* For the first statement, check that if at least two parameters are $0$, there is at least one $0$ in all $\binom{4}{3}$ subsets, making the whole regularizer $0$. If at least three parameters are non-zero, then there is at least one group that is non-zero. The second statement follows by symmetry. The third statement is valid because it is a cubic function when in the strict interior of an orthant. $\square$

*Proof of Proposition 2.* First observe that this regularizer applies pointwise to each coordinate of $W$. It suffices to prove the statements for one coordinate $w, w_0$. w.l.o.g. assume $w_0 > 0$, then the regularizer gives $\left|\frac{w(w-w_0)}{(1+\epsilon)w_0}\right|^2$. Observe that the nullspace is either $0$ or $w = w_0$, thus checking Statement 1. Statement 2 follows because all coordinates with $w = 0$ contributes $0$ to the total. Statement 3 follows because this is a fourth order polynomial of $w$, thus continuously differentiable. $\square$

*Proof of Proposition 6.* We start with $S = 4$. Define $f := g_4$ as a shorthand.

It should be noted that for all $1 \le i, j \le 4$, the partial functions $f_i(w_i) = f(w)$, for fixed $w_j$ with $j \ne i$, are strongly convex and quadratic. Therefore, when $i = 1$ for instance, we have for any $w_i$ and $v_i$ that

$$f_i(w_i) = f_i(v_i) + f_i'(v_i)(w_i - v_i) + \frac{1}{2}(w_i - v_i)^2.$$

Therefore, for any fixed $w_j$ with $j \ne i$, let $w_i^* = \arg\min_{w_i \ge 0} f_i(w)$. $w_i^*$ satisfies the following (when $i = 1$ for instance)

$$
\begin{aligned}
f(w) = f_1(w_1) &= f_1(w_1^*) + f_1'(w_1^*)(w_1 - w_1^*) + \frac{1}{2}(w_1 - w_1^*)^2 \\
&\ge f_1(w_1^*) + \frac{1}{2}(w_1 - w_1^*)^2 \\
&= f(w_1^*, w_2, w_3, w_4) + \frac{1}{2}(w_1 - w_1^*)^2.
\end{aligned}
\tag{8}
$$

The inequality is due to the first-order optimality condition. This means that we have a sufficient descent property with respect to each minimized variable.

**Proposition 8.** *Let $\{w^k\}_{k \in \mathbb{N}}$ be a sequence generated by the Alternating Minimization algorithm. Then, for all $k \in \mathbb{N}$, we have that*

$$f(w^k) \ge f(w^{k+1}) + \frac{1}{2}\|w^{k+1} - w^k\|^2.\tag{9}$$

*Proof.* Let $k \in \mathbb{N}$. Using (8) for all $1 \le i \le 4$ yields

$$
\begin{aligned}
f(w^k) &\ge f(w_1^{k+1}, w_2^k, w_3^k, w_4^k) + \frac{1}{2}(w_1^{k+1} - w_1^k)^2 \\
f(w_1^{k+1}, w_2^k, w_3^k, w_4^k) &\ge f(w_1^{k+1}, w_2^{k+1}, w_3^k, w_4^k) + \frac{1}{2}(w_2^{k+1} - w_2^k)^2 \\
f(w_1^{k+1}, w_2^{k+1}, w_3^k, w_4^k) &\ge f(w_1^{k+1}, w_2^{k+1}, w_3^{k+1}, w_4^k) + \frac{1}{2}(w_3^{k+1} - w_3^k)^2 \\
f(w_1^{k+1}, w_2^{k+1}, w_3^{k+1}, w_4^k) &\ge f(w^{k+1}) + \frac{1}{2}(w_4^{k+1} - w_4^k)^2.
\end{aligned}
$$

Adding all the inequalities yields the desired result. $\square$

Telescope (9), we get that

$$\min_{k \in [T]} \|w^{k+1} - w^k\|^2 \le \frac{1}{T}\sum_{k=1}^{K} \|w^{k+1} - w^k\|^2 \le \frac{f(w^0) - f(w^{k+1})}{T} \le \frac{4\lambda\|z\|^3}{T}.$$

The last inequality follows as we initialize at $w^0 = z$, thus $f(w^0) \le 4\lambda\|z\|^3$. Also, since $w^k \in \mathbb{R}^4$, $f(w^{k+1})$ is non-negative. This completes the proof for the first statement.

Now take $\epsilon \to 0$, as the position this algorithm halt, $\|w^{k+1} - w^k\|^2 \le \epsilon^2 \to 0$.

By Theorem 1 of (Beck et al., 2016), $w^k$ at $k \to \infty$ is a critical point of the objective function with the non-negative constraints handled by adding an indicator function.

The argument for the $S = 3$ case follows analogously (hence omitted). $\qquad\square$

*Proof of Proposition 7.* Under the assumption, the function $\text{Reg}_{W_0}$ has a locally Lipschitz continuous gradient, which implies that the function $f$ also has a locally Lipschitz continuous gradient. Therefore, the convergence of the sequence $\{x_t\}_{t\in\mathbb{N}}$ convergence to a critical point of the function $\psi \equiv f + h$ follows immediately from (Cohen et al., 2022) (since $\psi$ is a semi-algebraic function and $\{x_t\}_{t\in\mathbb{N}}$ is bounded).

$\qquad\square$

# B. Hyperparameters and configurations

Table 8 presents the configurations and hyperparameters used in our experiments. There are three key hyperparameters for learning an optimal semi-structured mask: sparsity regularization strength ($\lambda_1$), frozen weight regularization extent ($\lambda_2$), and learning rate. As discussed in Section 4.3.3, the frozen weight regularization is robust across a wide range of values. Our learning procedure follows standard settings, using *AdamW* as the optimizer with a warmup ratio of 0.1.

*Table 8.* Configure of the parammeter used in the experiment

|  | $\lambda_1$ | $\lambda_2$ | Learning rate | Optimizer | Warmup-ratio |
|---|---|---|---|---|---|
| Mistral-v0.1-7b | 20 | 0 | 5.00E-05 | Adamw | 0.1 |
| Mistral-v0.3-7b | 25 | 0 | 5.00E-05 | Adamw | 0.1 |
| Qwen-2.5-14b | 0.2 | 0 | 0.0001 | Adamw | 0.1 |
| OpenLlama-7b-v2 | 1 | 0 | 0.0001 | Adamw | 0.1 |
| Llama-2-7b | 0.25 | 0 | 0.0001 | Adamw | 0.1 |
| Llama-2-13b | 0.5 | 0.25 | 0.0001 | Adamw | 0.1 |
| Llama-3.1-8b | 0.85 | 0 | 5.00E-05 | Adamw | 0.1 |

# C. Regularization trajectory of the optimization algorithm.

We illustrate the regularization path for an example initialization with different $\lambda$ value using different optimization algorithms (EnumIPM, EnumPGD, and EnumALM) in Figure 3 as explained in Section 3.3.

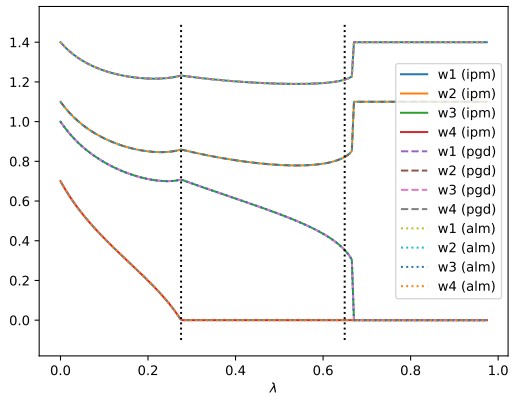

*Figure 3.* Illustration of the solution to (6) with an example input $y = [1.4, 1.1, 1.0, 0.7]$ as $\lambda$ increases. Observe that (1) the regularizer shrinks different coordinates differently according to their relative magnitude (2) all three algorithms return the same solution path. (3) the dashed lines indicate the two thresholds of $\lambda$ from KKT conditions above which the 3-sparse and 2-sparse solutions become critical points (a necessary condition for them to become global optimal).

*Table 9.* Experimental results on Wikitext perplexity (PPL) and performance across 7 commonly used zero-shot natural language reasoning tasks comparing **ProxSparse** to 3 other baselines on Llama-3.1-8b. **Bold** indicates the best pruning performance, while *italic* represents the original unpruned performance. SparseGPT updates weights to minimize reconstruction error, while the other methods keep retained weights frozen. Similar to the results in Table 2, ProxSparse consistently yields better results compared to all other baselines.

| | Weight Update | Wikitext PPL | ARC-C | ARC-E | SIQA | HellaSwag | OBQA | PIQA | TruthfulQA | AVG |
|---|---|---|---|---|---|---|---|---|---|---|
| Llama-3.1-8b | - | *5.84* | *0.515* | *0.814* | *0.470* | *0.600* | *0.334* | *0.801* | *0.368* | *0.557* |
| magnitude | ✗ | 766.91 | 0.257 | 0.454 | 0.365 | 0.335 | 0.154 | 0.634 | **0.319** | 0.360 |
| SparseGPT | ✓ | 14.61 | 0.316 | **0.647** | **0.426** | 0.435 | 0.222 | 0.705 | 0.301 | 0.434 |
| Wanda | ✗ | 20.91 | 0.269 | 0.573 | 0.400 | 0.380 | 0.192 | 0.686 | 0.309 | 0.401 |
| ProxSparse | ✗ | **13.63** | **0.333** | 0.623 | 0.422 | **0.460** | **0.240** | **0.721** | 0.296 | **0.444** |

*Table 10.* Experimental results on Wikitext perplexity (PPL) and 7 commonly used zero-shot natural language reasoning tasks comparing **ProxSparse** to 3 other baselines Llama-2-7b and Mistral-v0.3-7b model. **Bold** indicates the best pruning performance, while *italic* represents the original unpruned performance. AdmmPrune updates weights to minimize reconstruction error, while OWL and AlphaPrune uses dynamic sparse ratios across layers. ProxSparse consistently yields better results compared to all other baselines.

| | Weight Update | Wikitext PPL | ARC-C | ARC-E | SIQA | HellaSwag | OBQA | PIQA | TruthfulQA | AVG |
|---|---|---|---|---|---|---|---|---|---|---|
| Mistral-v0.3-7b | - | *4.95* | *0.490* | *0.797* | *0.458* | *0.609* | *0.336* | *0.803* | *0.353* | *0.549* |
| OWL | ✗ | 13.03 | 0.275 | 0.594 | 0.406 | 0.417 | 0.188 | 0.688 | 0.320 | 0.413 |
| AlphaPrune | ✗ | 13.58 | 0.265 | 0.529 | 0.398 | 0.407 | 0.190 | 0.668 | **0.335** | 0.399 |
| ADMMPrune | ✓ | 9.06 | 0.340 | 0.680 | 0.416 | 0.471 | 0.240 | 0.739 | 0.299 | 0.455 |
| ProxSparse | ✗ | **8.68** | **0.362** | **0.697** | **0.429** | **0.525** | 0.242 | **0.751** | 0.321 | **0.475** |
| Llama-2-7b | - | *5.12* | *0.433* | *0.763* | *0.461* | *0.571* | *0.314* | *0.781* | *0.321* | *0.521* |
| OWL | ✗ | 13.17 | 0.287 | 0.591 | 0.407 | 0.420 | 0.228 | 0.695 | **0.339** | 0.425 |
| AlphaPrune | ✗ | 13.01 | 0.293 | 0.607 | 0.406 | 0.411 | 0.238 | 0.690 | 0.317 | 0.424 |
| ADMMPrune | ✓ | 9.67 | 0.328 | 0.653 | **0.413** | 0.440 | **0.248** | 0.714 | 0.302 | 0.442 |
| ProxSparse | ✗ | **8.51** | **0.331** | **0.656** | 0.407 | **0.478** | 0.242 | **0.716** | 0.328 | **0.452** |

# D. End-to-end evaluation results on Llama-3.1-8b

In this section, we further discuss the evaluation results from Table 2, focusing on Llama-3.1-8b. We present results on Wikitext perplexity (PPL) and performance across seven commonly used zero-shot natural language reasoning tasks, comparing ProxSparse to three baselines in Table 9. In the Llama-3.1-8b experiments, ProxSparse significantly reduces perplexity from Wanda's 20.91 to 13.63. The overall results, compared to Magnitude Pruning, Wanda, and SparseGPT, follow the same trends discussed in Section 4.2, with ProxSparse consistently outperforming all other baselines.

# E. Comparison with Additional Pruning Baselines (ADMMPrune, OWL, and AlphaPrune)

In this section, we compare ProxSparse with three additional pruning baselines: ADMMPrune (Boža, 2024), OWL (Yin et al., 2023), and AlphaPrune (Lu et al., 2024). The experiments are done on Mistral-v0.3-7b and Llama-2-7b model. ADMMPrune introduces a fast and effective weight update algorithm for layerwise pruning based using the Alternating Direction Method of Multipliers (ADMM). As shown in table 10, ProxSparse outperforms ADMMPrune in both models, achieving lower PPL (8.51 vs. 9.67) and higher acc (47.6% vs. 45.5%), highlighting its effectiveness. We attribute the superority of ProxSparse to its end-to-end optimization process, which goes beyond solely relying on local layer-wised information.

OWL and AlphaPrune aim to determine layer-specfic ratio to protect important layers. Here we argue that they are not the best-suited mechanism in semi-structured pruning, as the sparse operator supported by hardware typically requires all blocks to strictly adhere the pattern, making applying varying ratios hard. Nevertheless, we conduct experiments on AlphaPrune and OWL for comparison. We follow mixed sparsity proposed in OWL and AlphaPrune with Wanda, that layers can have varying ratios, while the overall ratio remains 2:4. We see ProxSparse outperforms OWL and Alphaprune on Llama-2-7b and Mistral-v0.3-7b on PPL and accuracy, showing the strength of our end-to-end optimization. Further, as pruning patterns

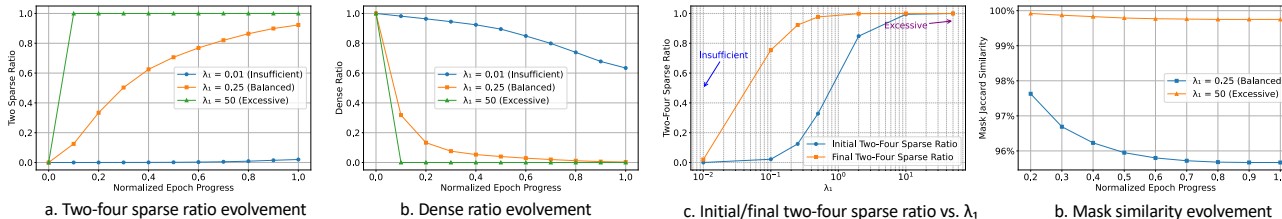

| a. Two-four sparse ratio evolvement | b. Dense ratio evolvement | c. Initial/final two-four sparse ratio vs. $\lambda_1$ | b. Mask similarity evolvement |

*Figure 4.* Evolution of sparsity ratio on OpenLlama-7b-v2 based on the degree of regularization. (a) Evolution of the 2:4 sparsity ratio over learning, where an insufficient regularization degree leads to under-learning. (b) With a larger $\lambda_1$ parameters shrink more quickly towards 2:4 sparsity, resulting in early commitment to a suboptimal mask. (c) Comparison of the 2:4 sparse block ratios at early (0.1 epochs) and final stages of learning. (d) Mask similarity between the final mask and the early mask obtained after 10% of learning. An excessively large $\lambda_1$ results in premature mask commitment, causing mask selection to stagnate and hindering optimal mask discovery.

become more fine-grained (e.g., 2:4), varying layer-wise pruning ratios become less effective as critical weights might still be removed within each block. This was reported in (Yin et al., 2023; Lu et al., 2024), where 4:8 pruning performed just similarly to uniform pruning in Wanda. This highlights the benefits of ProxSparse in identifying fine-grained semi-structured masks.

# F. Evolution of 2:4 sparsity across $\lambda_1$ on OpenLlama-7b-v2

In this section, we expand on the discussion from Section 4.3.2 and present the evolution trajectory of 2:4 sparsity across different $\lambda_1$ values on OpenLlama-7b-v2. Our findings on OpenLlama-7b-v2 are consistent with the main paper's discussion: a balanced regularization strength enables flexible mask exploration, preventing premature commitment while also facilitating more effective learning compared to excessively low values.

# G. Practical scenario of 2:4 sparsity and its extensibility discussion

To the best of our knowledge, commercially available hardware such as Nvidia Ampere GPUs, only supports the 2:4 sparsity pattern[1] . Our method directly aligns with the hardware features, making it directly applicable to real-world use cases. Meanwhile, our regularizer is flexible; extending to a 1:4 sparsity pattern is straightforward, as the regularizer can be reformulated and solved with even greater efficiency. On the otherhand, semi-structured patterns like 4:8 increase regularization terms, which could slow the solving process. Despite the longer but tolerable search time, inference gain remains unaffected once the optimal mask is found. A more efficient solver could further improve handling of such complex patterns, and we leave this for future exploration.

In the meantime, we note that increasing sparsity complexity (e.g., 2:4 to 4:8) will expand the search space, which is a common challenge for learning-based methods, including MaskLLM (Fang et al., 2024). Nevertheless, our regularizer supports extensibility and shows practical benefits in real-world scenarios.

---

[1]NVIDIA AMPERE GA102 GPU ARCHITECTURE Whitepaper

