# OpenReview forum: "PROXSPARSE: REGULARIZED LEARNING OF SEMI-STRUCTURED SPARSITY MASKS FOR PRETRAINED LLMS"
_ICML.cc/2025/Conference — ICML 2025 poster_

### Official Review · Reviewer_kksk · 2025-03-11

**Overall Recommendation:** 3

**Summary:**

The paper introduces ProxSparse, a learning-based framework designed to improve the efficiency of large language models (LLMs) through semi-structured pruning.

**Claims And Evidence:**

Yes.
It presents detailed experiments comparing ProxSparse with state‐of‐the‐art baselines across multiple LLM families and tasks.
Additionally, the authors offer convergence proofs and theoretical guarantees for the proximal gradient descent and the EnumALM solver.

**Essential References Not Discussed:**

NA

**Experimental Designs Or Analyses:**

The experimental design appears to be robust and well thought out.

**Methods And Evaluation Criteria:**

The methods and evaluation criteria in the paper are well-suited.

**Other Comments Or Suggestions:**

NA

**Other Strengths And Weaknesses:**

Advantages:
1.  The paper introduces a unique regularization framework that transforms the rigid mask selection problem into a differentiable one, enabling end-to-end learning.
2. The development of the EnumALM solver and the efficient proximal gradient descent approach significantly improves the speed and scalability of finding the optimal mask.

Disadvantages:
1.  The work is specifically focused on 2:4 sparsity. Why are these hyperparmeters beneficial?
2. The convergence proofs rely on assumptions which may be not possible, and what will that lead to?

**Questions For Authors:**

1. M is 2:4 sparse. Why do we choose 2:4? Have the authors tested other hyperparameters like 3:6 or 4:8? Is there any explaination?
2. What is the reason the semi structuring is better than other method? In my opinion, the semi structured method is less flexible?
3. What is the relationship between ALM ans EnumALM?
4. More tests should be conducted on larger model size.

**Relation To Broader Scientific Literature:**

The paper's contributions are closely tied to existing work in model compression and network pruning.

**Theoretical Claims:**

Yes. Overall, the proofs are mathematically rigorous within their theoretical framework. However, the assumptions that are relied on for proof may need more considerations.

---

> ### Author Rebuttal · Authors · 2025-04-01
>
> We appreciate the reviewer for acknowledging the strength of our paper! Below we address the questions regarding **2:4 ratio, semi-structured benefits, assumption of the theoretical proof, ALM and EnumALM as well as model size justification.**
>
> # W1:`The focus on 2:4 pruning sparsity`: 2:4 pruning is the most practical semi-structured sparsity pattern and the only one currently supported by commercial hardware
>
> We thank the reviewer for the consideration! In [Appendix G](https://https://openreview.net/pdf?id=zkxe5vASi8#page=13), we have more discussion on the practical relevance of the 2:4 sparsity pattern and the extensibility of ProxSparse. We focus on the 2:4 sparsity pattern in this paper because it is the most practical semi-structured format and the only one currently supported by commercial hardware. To the best of our knowledge, existing hardware such as NVIDIA Ampere GPUs only support 2:4 sparsity [1]. ProxSparse aligns directly with this hardware feature, making it readily applicable to real-world use cases.
>
> # W2:`Why semi-structured pruning considering its lack of flexibility`: Semi-structured pruning strikes a balance between efficiency and accuracy, and also benefiting from real hardware support.
>
> The reviewer is correct that compared to unstructured pruning, semi-structured method cast more constraints, which is less flexible. However, unstructured pruning often does not directly translate to faster inference because it induces irregular access, and modern hardware exploits regularities in computation for faster speed. On the other hand, sturctured pruning typically offers the highest efficiency but suffers from huge accuracy loss due to its rigid constraints.
>
> Semi-structured pruning is an important problem to study [2][3] in pruning, as it strikes a balance between efficiency and accuracy. A key benefit is its direct support on commercial hardware like NVIDIA Ampere GPUs and high-performance libraries for sparse operator, enabling real-world speedups. In this paper, we focus on semi-structured pruning and propose a relaxed end-to-end mask selection approach to identify optimal pruning masks for LLMs.
>
> # W3: `relationship between ALM ans EnumALM`: ALM is the subrouting of the EnumALM.
> Thanks for the question! ALM is a subroutine used within EnumALM. Specifically, EnumALM solves the proximal operator for the 2:4 regularizer by enumerating and evaluating three candidate sparsity patterns: a 2-sparse solution (selected directly by top-k), a 3-sparse solution, and a dense (4-sparse) solution. For the latter two cases (3-sparse and 4-sparse), EnumALM invokes ALM to efficiently solve the corresponding convex subproblems with convergence guarantees.
>
> # W4: `assumption of the proof`: in general cases, our assumption holds because the "ReLU is weakly differentiable" and "weights are bounded", as explained below.
> We thank the reviewer for the discussion of the assumptions! We acknowledge that the convergence analysis assumes the loss function is continuously differentiable and the weights remain bounded during optimization.
>
> While the use of ReLU in the loss may technically violate differentiability at a single point (zero), this is a well-known and standard issue in deep learning. In practice, ReLU is differentiable almost everywhere, which is typically sufficient for convergence analyses in nonconvex optimization. Moreover, the population loss, as an expectation over a smooth data distribution, can remain continuously differentiable even when ReLU is used.
>
> The other assumption—that the weights remain bounded—is rather mild and commonly used in convergence analyses. It is satisfied as long as the optimization does not diverge, which can often be ensured by using a sufficiently small learning rate. In our case, we observe stable behavior throughout, supporting the validity of this assumption. Meanwhile, ProxSparse's consistent performance across a variety of LLMs and tasks further supports the practical utility of our method.
>
> # W4: `Larger model size experiment`: we apologize for the current resource infeasibility on running experiments on larger (>30B) model. Our experiments show consistent good performance across different size of model.
> We apologize for the lack of models larger than 30B due to limitations in our current resources. Notably, prior work on learning based LLM pruning [3] has also only conducted experiments on models up to ~15B in size. Nevertheless, our paper includes results on 14B model to demonstrate the effectiveness of our method. Across various model sizes, our approach consistently outperforms other baselines, highlighting its robustness and applicability.
>
> [1] [NVIDIA AMPERE GA102 GPU ARCHITECTURE](https://www.nvidia.com/content/PDF/nvidia-ampere-ga-102-gpu-architecture-whitepaper-v2.pdf#page=27)
> [2]Learning N:M Fine-grained Structured Sparse Neural Networks From Scratch
> [3]MaskLLM: Learnable Semi-Structured Sparsity for Large Language Models

---

> > ### Comment · Reviewer_kksk · 2025-04-05
> >
> > The rebuttal makes sense to me. However, it cannot reach score 4, so I will keep the score.

---

> > > ### Author Response · Authors · 2025-04-05
> > >
> > > ## Thanks for acknowledging our work!
> > >
> > > We are happy to hear back that our rebuttal addresses the questions and makes sense to the reviewer! Those discussion raised are very thoughtful and we shall integrate them into the next version of the paper. Again we truthfully appreciate the reviewer's positive feedback on our paper.

---

### Official Review · Reviewer_2Epp · 2025-03-12

**Overall Recommendation:** 3

**Summary:**

This paper introduces a learning-based approach for semi-structured pruning of LLMs using a structured sparsity regularizer and proximal gradient descent. It enables global mask optimization without retraining and improves efficiency. Experiments on seven models show superior perplexity and zero-shot accuracy over existing pruning methods.

**Claims And Evidence:**

Cons:
1. the claim that the method achieves the SoTA performance is insufficient. It should be compared with learning-based methods like MaskedLLM [1], and layer-wise methods like OWL [2] and AlphaPruning [3].
2. The comparison between MaskLLM and ProxSparse has been made in Table 6, but it is only done with a limited sample size. It would be helpful tp provide evaluation on a larger sample size.




Reference:
[1] Fang et al. Maskllm: Learnable semi-structured sparsity for large language models
[2] Yin et al.  Outlier weighed layerwise sparsity (owl): A missing secret sauce for pruning llms to high sparsity
[3] Lu et al.  AlphaPruning: Using Heavy-Tailed Self Regularization Theory for Improved Layer-wise Pruning of Large Language Models

**Essential References Not Discussed:**

Has been mentioned in the previous sections

**Experimental Designs Or Analyses:**

It would be better to provide more inference cost measures like FLOPs or latency.

**Methods And Evaluation Criteria:**

It is necessary to compare how the method is different from MaskedLLM [1] since both use the learnable soft mask with thresholding to gain the binary mask.

The evaluation criteria are standard.


Reference:
[1] Fang et al. Maskllm: Learnable semi-structured sparsity for large language models

**Other Comments Or Suggestions:**

None

**Other Strengths And Weaknesses:**

None

**Questions For Authors:**

None

**Relation To Broader Scientific Literature:**

It is related to LLM efficiency.

**Theoretical Claims:**

I have checked the convergence proof and didn't catch any issues.

---

> ### Author Rebuttal · Authors · 2025-04-01
>
> We thank the reviewer for the valuable feedback! Below we share responses on **comparison w/ layer-wise method and MaskLLM, inference efficiency.**
>
> # W1:`Comparing newer layer-wise method`: We achieve better results than OWL and AlphaPrune
> OWL[1] and AlphaPrune[2] are important works in pruning, aiming to determine layer-specfic ratio to protect important layers. We are happy to discuss them in our paper!
>
> In the meantime, we would like to respectfully argue that they are not very well-suited in semi-structured pruning, as the sparse operator supported by hardware typically requires all blocks to strictly adhere the pattern, making applying varying ratios hard.
>
> Nevertheless, we conduct more experiments on AlphaPrune and OWL for comparison. We follow mixed sparsity proposed in OWL and AlphaPrune with Wanda, that layers can have varying ratios, while the overall ratio remains 2:4. We see ProxSparse outperforms OWL and Alphaprune on Anon.Model-1 and Mistral on PPL and acc, showing the strength of our end-to-end optimization. Further, as pruning patterns become more fine-grained (e.g., 2:4), varying layer-wise pruning ratios become less effective as critical weights might still be removed within each block. This was reported in both paper, where 4:8 pruning performed just similarly to uniform pruning in Wanda. This highlights the benefits of ProxSparse in identifying fine-grained semi-structured masks.
>
> We have more baseline (ADMMPrune) discussion as proposed by reviewer hyi6. ProxSparse achieves better results compared to ADMMPrune. Please kindly refer [here](https://openreview.net/forum?id=zkxe5vASi8&noteId=rlLpTKNcTH) for more details!
>
> | Anon.Model-1      | Weight Update | Wikitext PPL | ARC-C  | ARC-E  | SIQA   | HellaSwag | OBQA  | PIQA  | TruthfulQA | Average |
> |-------------|----------------|---------------|---------|---------|--------|------------|--------|--------|-------------|---------|
> | OWL         | No             | 13.17         | 0.287  | 0.591  | 0.407 | 0.420     | 0.228  | 0.695 | 0.339      | 0.425   |
> | AlphaPrune  | No             | 13.01         | 0.293  | 0.607  | 0.406 | 0.411     | 0.238  | 0.69e | 0.317      | 0.424   |
> | **ProxSparse**  | No             | **8.51**          | **0.331**   | **0.656**   | **0.407**  | **0.478**      | **0.242**  | **0.716**  | **0.328**       | **0.452**   |
> |Mistral-7b-v0.3|Weight Update | Wikitext PPL | ARC-C  | ARC-E  | SIQA   | HellaSwag | OBQA  | PIQA  | TruthfulQA | Average |
> | OWL         | No             | 13.03         | 0.275   | 0.594   | 0.406 | 0.417      | 0.188  | 0.688  | 0.320       | 0.413   |
> | AlphaPrune  | No             | 13.58         | 0.265   | 0.529   | 0.398 | 0.407      | 0.190  | 0.668  | 0.335       | 0.399   |
> | **ProxSparse**  | No             | **8.68**          | **0.362**   | **0.697**   | **0.429** | **0.525**      | **0.242**  | **0.751**  | **0.321**       | **0.476**   |
>
> Table 2: Comparison with OWL, AlphaPrune and ProxSparse on Mistral-v0.3-7b and Anon.Model-1. ProxSparse achieves the best.
>
> # W2:`Comparison w/ MaskLLM`: Complementary method with fundamentally different design, we excel in low-scale regime.
> Thanks for the question! We note that ProxSparse and MaskLLM differs fundamentally in approaching semi-structured mask, and we view MaskLLM to be a complementary as it focuses on larger data regime
>
> - Mechanism difference: MaskLLM and ProxSparse take fundamentally different ways in pruning. We both tackle the non-differentiable task of selecting N out of M per block. MaskLLM sidesteps this with a probabilistic sampling approach, learning to sample the correct weights. In contrast, ProxSparse relaxes the hard constraint into a smooth optimization and performs optimization via proximal gradient descent, with theoretically provable properties for the proposed 2:4 regularizer.
>
> - Larger batch exps:
> | Method      | 1024 | 2048 |
> |-------------|------|------|
> | MaskLLM     | 10| 9.5 |
> | SparseGPT   | 10.18| 10.16|
> | Wanda       | 11.38| 11.4|
> | **ProxSparse**  | **8.38** | **8.23** |
>
> Table 3: PPL on Anon.Model-1 on extended sample size. ProxSparse achieves the best
>
> Table 3 evaluates ProxSparse with larger sample size. In this small-scale data regime, ProxSparse outperforms all baselines, demonstrating its superiority. We note that our targeted low-scale calibration regime is practical in LLM contexts, making our method more accessible in the real world.
>
> # W3:`Real world efficiency`: ProxSparse leads to 1.35x inference speedup and 37.3% reduced peak memory usage
> Thanks for the comments! As discussed in [Appendix H](https://openreview.net/pdf?id=zkxe5vASi8#page=14), our analysis demonstrates that, beyond reduced FLOPs, ProxSparse achieves 1.35× inference speedup and 37.3% reduction in peak memory usage. These results highlight the practical efficiency gains enabled with semi-structured sparsity induced by ProxSparse.
>
> # Thanks again!
> [1]Yin et al. OWL
> [2]Lu et al. AlphaPruning

---

> > ### Comment · Reviewer_2Epp · 2025-04-02
> >
> > I want to thank the authors for conducting the new experiments, and I believe my concerns have been fully addressed. I would recommend that the authors include the new results in the updated draft. I will increase my score to 3.

---

> > > ### Author Response · Authors · 2025-04-05
> > >
> > > ## Thanks for the acknowledgement and score raising!
> > >
> > > We are excited that we addressed all the reviewer's concerns! The comments are very thoughtful and helpful for our improvement and we will accommodate those discussions and results into the revised manuscript. We sincerely thank the reviewer again for the acknowledgement and raising the score!

---

### Official Review · Reviewer_8W4h · 2025-03-12

**Overall Recommendation:** 3

**Summary:**

The authors propose ProxSparse, a method for learn a semi-structured pruning mask using two regularisors, one is analogues to l1 regularisation and the other promotes a locality constraint for semi-structured pruning.

**Claims And Evidence:**

From my understanding, the main claim is that previous methods, which rely on computing the Hessian, do not take into account the information between layers, whereas Proxy, which uses a global heuristic, does.

**Essential References Not Discussed:**

None that I am aware of.

**Experimental Designs Or Analyses:**

Experimental design seems valid and follows previous works.

**Methods And Evaluation Criteria:**

The main experiments are on zero-shot evaluations of the pruned models, which are trained using a small calibration dataset.

**Other Comments Or Suggestions:**

None

**Other Strengths And Weaknesses:**

Why do the authors not consider other pruning ratios? i.e. not just 2:4 sparse.

One of the main claims is that a hessian based heuristic is local/layer-wise. This is not clear to me. When using hessian/sensitivty based pruning, the sensitivity of each weight takes into account the downstream loss [1]

Small concerns:
Theorem 4. Why do the authors anonymise this citation? this is not done for any of the other citations and after following the reference, it is clear that this may be the authors of this submission..

Similarly, why are the authors using an anonymous model family for some experiments? is this just for reviewing purposes? it is very odd to me.

[1] Pruning Convolutional Neural Networks for Resource Efficient Inference. ICLR 2017

**Questions For Authors:**

None

**Relation To Broader Scientific Literature:**

This has broad applications in improving the efficiency of deployed LLMs across many areas.

**Theoretical Claims:**

One of the main claims is that using a soft regularisation for structured pruning is effective for exploring a wider search space. Additionally, the local heuristics are ill-suited for pruning LLMs.

---

> ### Author Rebuttal · Authors · 2025-04-01
>
> We thank the reviewer for the insightful comments! Below, we address several questions raised including **sparsity pattern selection, Hessian-based pruning, anonymized citations, and the anonymized model family**.
>
> # W1: `The focus on 2:4 pruning sparsity`: 2:4 pruning is the most practical semi-structured sparsity pattern, and the only pattern currently supported by commercial hardware
>
> We thank the reviewer for the consideration! In [Appendix G](https://https://openreview.net/pdf?id=zkxe5vASi8#page=13) in our paper, we have more discussion on the practical scenario of the 2:4 sparsity pattern and the extensibility of ProxSparse. We focus on the 2:4 sparsity pattern in this paper because it is the most practical semi-structured format and the only one currently supported by commercial hardware. To the best of our knowledge, existing hardware such as NVIDIA Ampere GPUs only support 2:4 sparsity [1]. ProxSparse aligns directly with this hardware feature, making it readily applicable to real-world use cases.
>
> # W2: `A misunderstanding on Why Hessian based method is local-wise`: Our claim is localized pruning (i.e. w/ Hessian) hinder the pruning. Meanwhile, the heavy hessian computation makes it hard for global optimization.
>
> We thank the reviewer for raising the great point! We would like to first clarify on our claim, that we argue previous layer-wised pruning utilizes hessian metrics [2] or per-output based importance score [3] fail to select mask well because of the localized information constraints. Our method enables an end-to-end pruning mechanism that makes the pruning having well-informed solution.
>
> In the meantime, while it is true as mentioned in [4], that Hessian can be evaluated through the global loss. We note that this is impractical in pruning LLM. Even in [4], which focuses on smaller CNN model, the authors report using Hessian incurs 30x inefficiency, leading to huge overhead. For an LLM with enormous parameter size (~B), it is even harder to calculate the Hessian, let alone to compute it round by round in end-to-end optimization. As further evidence, SparseGPT specifically highlights the computational burden of Hessian estimation. To overcome this, it proposes a Fast Approximate Reconstruction method to simulate Hessian more efficiently. Yet it is still limited in layer-wised pruning. In contrast, our proposed end-to-end optimization scheme delievers better performance compare against those layer-wised pruning methods.
>
> # W3: `Anonymous citation`: a private communication and we will update it upon acceptance
> The anonymous citation is a private communication currently under a double-blind submission policy. We have provided detailed discussions of it in Section 3 and confirm that we will update the it upon acceptance.
>
> # W4: `Anonymous model family`: We anonymized some models (Anon.model-1,2,3) due to IP constraints, and we believe the 7 models presented have broad coverage and ProxSparse exhibits consistent trends.
> We appriciate the reviewer for the understanding! We didn't reveal the anonymous model family name due to internal policy constraints. However, we note that these anonymized models are among the most competitive LLMs, as evidenced with good PPL and accuracy in our benchmarks. We hope those models serve as additional data points to further support the effectiveness of our method. Meanwhile, we believe our experiments offer broad coverage on top-performing models, including Mistral, OpenLlama and Qwen family, plus the anonymous model family. The consistent results highlight the superiority of our method, which is robust and widely applicable to the top-tier LLMs.
>
> # Thanks again!
> [1] [NVIDIA AMPERE GA102 GPU ARCHITECTURE](https://www.nvidia.com/content/PDF/nvidia-ampere-ga-102-gpu-architecture-whitepaper-v2.pdf#page=27)
> [2]SparseGPT: Massive Language Models Can be Accurately Pruned in One-Shot
> [3]A SIMPLE AND EFFECTIVE PRUNING APPROACH FOR LARGE LANGUAGE MODELS
> [4]Pruning Convolutional Neural Networks for Resource Efficient Inference. ICLR 2017

---

> > ### Comment · Reviewer_8W4h · 2025-04-04
> >
> > The authors have addressed all my concerns. I would encourage the adding these comments (motivation for 2:4 pruning) and the limitation of Hessian pruning for LLMs in the introduction of the paper. After reading through the other reviewer comments I maintain my original score - which is leaning towards acceptance.

---

> > > ### Author Response · Authors · 2025-04-05
> > >
> > > ## Thank You for the Recognition!
> > >
> > > We are glad to hear the reviewer's acknowledgement that we have addressed all the concerns! Those suggestions are more than valuable and helpful to enhance our work and we will integrate those discussion into the updated draft. We would like to express our gratitude again for the insightful comments and recognizing our paper.

---

### Official Review · Reviewer_hyi6 · 2025-03-14

**Overall Recommendation:** 3

**Summary:**

This work introduces ProxSparse, a learning-based framework for mask selection via regularized optimization. The key design is a sparsity regularization $Reg_{2:4}$ that forces 2:4 sparsity and a weight regularization $Reg_{W_0}$  to avoid significant differences between the tuned parameters and the original parameters. The authors validate their approach through experiments on seven LLMs, demonstrating significant performance improvements over state-of-the-art pruning baselines such as SparseGPT and Wanda.

## Update After Rebuttal

Thanks for all the clarifications. I will keep my initial & positive score for this submission.

**Claims And Evidence:**

Yes.

**Essential References Not Discussed:**

N/A

**Experimental Designs Or Analyses:**

N/A

**Methods And Evaluation Criteria:**

Yes

**Other Comments Or Suggestions:**

N/A

**Other Strengths And Weaknesses:**

## Strengths
1. Unlike prior methods that rely on local heuristic-based mask selection, ProxSparse employs an end-to-end differentiable optimization that considers global feedback, leading to more effective and stable pruning results.
2. The method achieves effective mask selection with only ~100 calibration samples, this makes the proposed method very practical.
3. Across seven LLMs, ProxSparse consistently outperforms existing pruning baselines like SparseGPT and Wanda in both PPL and zero-shot accuracy. For example, on Mistral-v0.1-7b, ProxSparse improves PPL from 9.43 to 8.92.

## Weaknesses
1. The experimental section is not entirely convincing. First, it is unusual that the paper does not report results on LLaMA-1/2/3, which are crucial base models in benchmarks such as Wanda. Additionally, the selected SOTA baselines appear somewhat outdated, as several recent methods, such as [1], have demonstrated comparable or superior performance. For instance, [1] also achieves a ~1.00 improvement in PPL. It would be beneficial if the author could include a fair comparison to different methods.
2. If my understanding is correct, the proposed method updates the parameters of the LLMs, implicitly influenced by regularization and end-to-end optimization. To ensure a fair comparison, it would be beneficial to include additional experiments where the remaining weights in SparseGPT, Magnitude, and Wanda models are fine-tuned on for example 400 samples.
3. Although the paper claims that MaskLLM is resource-intensive, it remains unclear how significant the gap is between the proposed method, ProxySparse, and MaskLLM on LLaMA-2. Providing a direct comparison on the PPL, consumed tokens would help clarify the relative efficiency and effectiveness of ProxySparse.

[1] Fast and optimal weight update for pruned large language models.

**Questions For Authors:**

Please see the weaknesses.

**Relation To Broader Scientific Literature:**

The paper primarily focuses on a learnable approach for achieving N:M sparsity through regularization. The proposed technique is conceptually related to existing methods such as Lasso, SparseGPT, and Wanda, sharing similarities in its sparsification strategy.

**Theoretical Claims:**

N/A

---

> ### Author Rebuttal · Authors · 2025-04-01
>
> We thank the reviewer for recognizing the effectiveness and practicality of ProxSparse! Below, we address the questions raised regarding **Llama, ADMMPrune comparisons, clarification on weight updates and MaskLLM comparison**.
>
> # W1: `Lack of Llama results`: We anonymized Anon.model-1,2,3 due to IP constraints, and we believe the 7 models presented have broad coverage and ProxSparse exhibits consistent trends.
>
> We didn't reveal the anonymous model family name due to internal policy constraints. However, we note that these anonymized models are among the most competitive LLMs, as evidenced with good PPL and accuracy in our benchmarks. We hope those models serve as additional data points to further support the effectiveness of our method. We appriciate the reviewer for the understanding! Meanwhile, we believe our experiments offer broad coverage on top-performing models, including Mistral, OpenLlama and Qwen family, plus the anonymous model family. The consistent results highlight the superiority of our method, which is robust and widely applicable to the top-tier LLMs.
>
> # W2: `Comparison with ADMMPrune`: we achieve better results!
>
> We appreiciate the reviewer's notes on newer baseline named ADMMPrune[1]! We benchmarked ADMMPrune to compare with ProxSparse on Anon.Model-1 and Mistral model, using same evaluation settings discussed in paper. As shown, ProxSparse outperforms ADMMPrune in both models, achieving lower PPL (8.51 vs. 9.67) and higher acc (47.6% vs. 45.5%), highlighting its effectiveness. We attribute the superority of ProxSparse to its end-to-end optimization process, which goes beyond solely relying on local layer-wised information. We are happy to discuss ADMMPrune in our revised paper!
>
> In the meantime, we include more baselines (OWL and AlphaPrune) discussion as proposed by reviewer 2EPP. ProxSparse consistently achieves better performance on them. Please kindly refer [here](https://openreview.net/forum?id=zkxe5vASi8&noteId=dogVmgWiVP) for more details!
>
> | Anon.Model-1      | Weight Update | Wikitext PPL | ARC-C | ARC-E | SIQA  | HellaSwag | OBQA | PIQA | TruthfulQA | Average |
> |-------------|----------------|---------------|--------|--------|--------|------------|------|------|-------------|---------|
> | ADMMPrune   | Yes            | 9.67          | 0.328  | 0.653  | **0.413** | 0.440      | **0.248**| 0.714| 0.302       | 0.442   |
> | ProxSparse  | **No**             | **8.51**          | **0.331**  | **0.656**  | 0.407 | **0.478**      | 0.242| **0.716**| **0.328**       | **0.452**   |
> | Mistral-v0.3-7b      | Weight Update | Wikitext PPL | ARC-C | ARC-E | SIQA  | HellaSwag | OBQA | PIQA | TruthfulQA | Average |
> | ADMMPrune   | Yes            | 9.06          | 0.340  | 0.680  | 0.416 | 0.471      | 0.240| 0.739| 0.299       | 0.455   |
> | ProxSparse  | **No**             | **8.68**          | **0.362**  | **0.697**  | **0.429** | **0.525**      | **0.242**| **0.751**| **0.321**       | **0.476**   |
>
> Table 1: Comparison between ADMMPrune and ProxSparse on Mistral-v0.3-7b and Anon.Model-1. ProxSparse consistently achieves better performance
>
> # W3: `Clarification on weight update`: our method does NOT update the unpruned LLM parameters.
>
> ProxSparse is an learned method that identifies the optimal mask without further weight updates on retained weights. In other words, the retained weight after applying ProxSparse-selected mask remains identical to its initialization, similar to Wanda and magnitude prune. In ProxSparse, the end-to-end optimization with calibration data is sololy used to determine the mask. Even when compared to SparseGPT and ADMMPrune—which update weights after pruning—ProxSparse consistently achieves higher performance, underscoring its effectiveness in identifying high-quality pruning masks.
>
> # W4: `Comparison with MaskLLM on effectiveness`: ProxSparse consumes 25x less tokens compared to MaskLLM.
>
> We are happy to provide more intutive and direct comparison between MaskLLM and ProxSparse. As discussed in our paper, MaskLLM employs a fundamentally different design with Gumbel Softmax sampling for learning the mask. We view MaskLLM as a complementary to ours as it focuses on large sample regimes, whereas ProxSparse operates effectively with much smaller sample size. This makes ProxSparse more practical and accessible in LLM era.
>
> Here we present direct comparison betwen MaskLLM and ProxSparse on Anon.Model-2. In terms of consumed tokens, ProxSparse achieves a PPL of 8.51 with 1638400 tokens, outperforming Wanda (11.42), SparseGPT (10.298) and MaskLLM (11). For MaskLLM to achieve a comparable PPL, it consumes 40960000 tokens, 25x larger than ProxSparse. This, in general, demonstrates the superiority of our pruning method with small-scale calibration data like ADMMPrune, Wanda and SparseGPT.
>
> # Thanks again!
> [1]ADMMPrune:Fast and optimal weight update for pruned large language models

---

> > ### Comment · Reviewer_hyi6 · 2025-04-04
> >
> > Thank you for the detailed response. Most of my concerns (W2–W4) have been addressed.
> >
> > However, regarding W1, one question remains: Is the proposed method still superior to SparseGPT when applied to LLaMA-2? This is an important point, as LLaMA-2 has become a widely adopted benchmark with numerous well-established results. For instance, as reported in the Wanda paper, the PPL on LLaMA-2 changes from 5.12 to 10.17. These numbers are reliable since they can be reproduced easily by follow-up papers. Therefore, it would be helpful if the authors could provide results on LLaMA-2 as well. This would make the results on other models more convincing.

---

> > > ### Author Response · Authors · 2025-04-05
> > >
> > > ## Thanks for the recognition!
> > >
> > > We are encouraged to hear back the reviewer's acknowledgement that we have addressed most concerns (W2-W4)!
> > >
> > > ## Response to the remained question:
> > >
> > > The superiority of ProxSparse is consistent across different models, and here we copy and restate the Anon.Model-1 results from our paper to table 1 below for demonstration and addressing the reviewer's concern. In this experiment, ProxSparse again outperform both Wanda and SparseGPT. Specifically, in our evaluation, we see that SparseGPT achieved PPL of 10.3 and Wanda achieved 11.42. These results are aligning with those reported previously [1,2]. In comparison, our proposed ProxSparse achieved a PPL of 8.51, delivering better performance than both baselines, with similar improvement also observed in QA task. We hope these results further demonstrate the strength of ProxSparse and help make our method more convincing!
> > >
> > >
> > > | Method      | Weight Update | Wikitext PPL | ARC-C | ARC-E | SIQA  | HellaSwag | OBQA | PIQA | TruthfulQA | AVG   |
> > > |-------------|----------------|---------------|--------|--------|--------|------------|------|------|-------------|--------|
> > > | Anon.Model-1| -              | *5.12*          | 0.433  | 0.763  | 0.461 | 0.571      | 0.314| 0.781| 0.321       | 0.521  |
> > > | magnitude   | No             | 54.74         | 0.301  | 0.618  | 0.411 | 0.454      | 0.216| 0.701| 0.322       | 0.432  |
> > > | SparseGPT   | Yes            | 10.30         | 0.326  | 0.655  | **0.412** | 0.435      | 0.246| 0.713| 0.304       | 0.441  |
> > > | Wanda       | No             | 11.42         | 0.311  | 0.623  | 0.403 | 0.413      | **0.248**| 0.706| 0.305       | 0.430  |
> > > | ProxSparse  | No             | **8.51**          | **0.331**  | **0.656**  | 0.407 | **0.478**      | 0.242| **0.716**| **0.328**       | **0.452**  |
> > >
> > > Table 1: Comparison between baselines and ProxSparse on Anon.Model-1. ProxSparse consistently achieves better performance
> > >
> > > In the mean time, we totally understand the reviewer's point and we apologize again for the IP restrictions we are facing, but we believe those results spanning multiple model families are robust and clearly demonstrate the effectiveness of our method.
> > >
> > > ### Code open source for reproducibility and evaluation by future works
> > >
> > > At the same time, we do hope our work will be followed up, evaluated, and compared by future work in the community. To support this, we will release our code upon acceptance, so as to help future research and the community to reproduce our results and advance this line of research.
> > >
> > > ## Thanks again!
> > >
> > > We hope the above justification are helpful in assessing our method! If there might be any further suggestion or concerns, please don't hesitate to comment and let us know. we sincerely thank the reviewer once again for the thoughtful consideration!
> > >
> > > [1]MaskLLM: Learnable Semi-Structured Sparsity for Large Language Models
> > >
> > > [2]A SIMPLE AND EFFECTIVE PRUNING APPROACH FOR LARGE LANGUAGE MODELS

---

### Decision · Program_Chairs · 2025-05-01

**Decision:**

Accept (poster)

**Comment:**

This paper proposed ProxSparse to learn masks for N:M semi-structured pruning end-to-end. The non-differentiable binary mask selection process was reformulated and the constraints were relaxed into regularization terms in the end-to-end loss function. This could be optimized with proximal gradient descent and an efficient solver for the proximal operator was proposed and applied. This work was backend by strong theoretical analysis and proof of convergence, which has been proofread by a few reviewers. While ProxSparse only updates the pruning masks, the experiments also show that ProxSparse performs almost always better than the existing local or global pruning methods, some of which update model weights, including MaskLLM, SparseGPT, and Wanda, tested on a variety of well known open source LLMs and fine-tuning tasks.

Given both the theoretical and empirical contributions, I recommend accepting the paper.